# A Quadruplex RT-qPCR for the Detection of Porcine Sapelovirus, Porcine Kobuvirus, Porcine Teschovirus, and Porcine Enterovirus G

**DOI:** 10.3390/ani15071008

**Published:** 2025-03-31

**Authors:** Biao Li, Kaichuang Shi, Yuwen Shi, Shuping Feng, Yanwen Yin, Wenjun Lu, Feng Long, Zuzhang Wei, Yingyi Wei

**Affiliations:** 1Guangxi Key Laboratory of Animal Breeding, Disease Control and Prevention, College of Animal Science and Technology, Guangxi University, Nanning 530005, China; libiao6096@126.com (B.L.); shiyuwen2@126.com (Y.S.); zuzhangwei@gxu.edu.cn (Z.W.); 2Guangxi Center for Animal Disease Control and Prevention, Nanning 530001, China; fsp166@163.com (S.F.); yanwen0349@126.com (Y.Y.); nnlwj@126.com (W.L.); longfeng1136@163.com (F.L.)

**Keywords:** porcine sapelovirus (PSV), porcine kobuvirus (PKV), porcine teschovirus (PTV), porcine enterovirus G (EV-G), multiplex RT-qPCR, co-infection

## Abstract

Porcine sapelovirus (PSV), porcine kobuvirus (PKV), porcine teschovirus (PTV), and porcine enterovirus G (EV-G) are important viruses in the pig industry. These viruses play important roles in the establishment of similar clinical signs of diarrhea, encephalitis, and reproductive and respiratory disorders in pig herds. The differential detection of these viruses is crucial for the accurate diagnosis of these diseases. In this study, a quadruplex real-time quantitative RT-PCR (RT-qPCR) for the simultaneous detection of PSV, PKV, PTV, and EV-G was developed. The assay showed high sensitivity, strong specificity, and excellent repeatability. The assay was further assessed for its applicability through testing 1823 fecal samples collected from different pig farms. The positivity rates of PSV, PKV, PTV, and EV-G were 15.25% (278/1823), 21.72% (396/1823), 18.82% (343/1823), and 27.10% (494/1823), respectively, with coincidence rates higher than 99.01% for the reference RT-qPCR/RT-PCR. The results indicated that the developed assay provides a rapid, sensitive, and accurate method for the simultaneous detection of PSV, PKV, PTV, and EV-G.

## 1. Introduction

Porcine sapelovirus (PSV) is classified in the *Sapelovirus* genus within the *Picornaviridae* family [1]. It is a positive-stranded RNA virus with a genome of 7.5 kb [2]. PSV can be isolated from the feces of healthy and diarrheic pigs [3]. The virus can cause severe diarrhea, respiratory distress, neurological symptoms, and reproductive failure in pigs [4]. The pathological changes such as atrophy of the villi are mainly found in the small intestine. Although, the virus is also detected in other organs, no significant pathological changes are found in these organs [5]. A recent study showed that PSV could infect human 293T cells, suggesting that PSV may have a potential risk of infection in humans [6]. PSV has been reported worldwide [2,3,4,5,7], including China [5,6,8,9].

Porcine kobuvirus (PKV) belongs to the *Kobuvirus* genus of the *Picornaviridae* family, and it is a single-stranded positive-sense RNA virus with a 8.1 kb genome [1,10]. Kobuvirus has been found in different animal species and humans around the world, and can cause diarrhea in both humans and animals, indicating the zoonotic potential of this pathogen [11,12]. Although PKV is an enterovirus, it can be present in the intestinal tissues with diarrhea or diarrheic subclinical pigs, and it can also induce other clinical signs such as respiratory distress [13]. A study found that the PKV-positive pigs were clinically healthy, while many pigs infected with PKV and porcine epidemic diarrhea virus (PEDV) showed clinical illness of diarrhea [13]. It has been shown that PKV can synergize with PEDV, and can exacerbate clinical signs and pathological damage to the infected pigs [14].

Porcine teschovirus (PTV) is a member of the *Teschovirus* genus within the *Picornaviridae* family. The genome of the virus is a single-stranded positive-sense RNA, 7.1 kb in size [1]. PTV-infected pigs usually show diarrhea or subclinical diarrhea, and other signs such as polioencephalomyelitis, reproductive disorders, and severe-to-moderate neurological symptoms [15,16,17]. PTV consists of 14 genotypes and may cause different damage to pigs depending on the viral genotypes. For example, PTV-1 is usually considered a genotype of porcine diarrhea and neurotropic signs [16,17], PTV-2 induces severe encephalomyelitis [18], and PTV-13 causes neurological symptoms in piglets [19]. In addition, some other genotypes of PTV have also been detected [20,21], but their pathogenic roles in pig herds require further investigation and evaluation.

Porcine enterovirus G (EV-G) belongs to the *Enterovirus* genus within the *Picornaviridae* family, and is a single-stranded positive-sense RNA virus with a 7.4 kb genome size [1]. Twenty genotypes of EV-G have been identified [22]. The first isolated genotype 2 EV-G, which can cause mild diarrhea, wasting, and other clinical signs in newborn piglets, was reported in Sichuan Province of China in 2023 [23]. In addition, EV-G can also induce neurological symptoms, and growth retardation [24]. It has been reported that there is extensive recombination between different genotypes of EV-G [25,26], and recombinant EV-G strains can persist in pig farms [27].

All four viruses, PSV, PKV, PTV, and EV-G, play important roles in the establishment of diarrhea in pigs, and PSV and PKV have the potential risk of infecting humans. The pigs infected with these four viruses may be subclinical or have similar signs of diarrhea and neurological symptoms. Co-infections of these viruses exacerbate the clinical signs and pathological changes of the diseases [6,13,14,28], which seriously jeopardize the health of pigs. It is difficult to distinguish these diseases relying on only the clinical signs and pathological changes, especially the existence of co-infections with these viruses. To date, there is no commercial kit to detect these four viruses in China. A multiplex RT-PCR assay was developed for the detection of PTV, PSV, and EV-G [29], but RT-PCR has shortcomings such as cumbersome operation and non-specificity of amplification. Therefore, multiplex real-time fluorescence quantitative RT-PCR (RT-qPCR) has gradually replaced multiplex RT-PCR, and become the major technique for the simultaneous detection of several viruses in clinical samples. Multiplex RT-qPCR can simultaneously detect multiple pathogens in the same sample in a short time, with the advantages of easy operation, strong specificity, and high sensitivity [30,31]. To date, no detection method for the simultaneous detection of PSV, PKV, PTV, and EV-G has been reported. In this study, a quadruplex RT-qPCR assay for the detection of PSV, PKV, PTV, and EV-G was developed, which could be used for the clinical detection and epidemiological investigation of these pathogens.

## 2. Materials and Methods

### 2.1. Reference Strains

The following vaccine strains were purchased from Keqian Biological Co., Ltd. (Wuhan, China): transmissible gastroenteritis virus (TGEV, H strain), porcine epidemic diarrhea virus (PEDV, CV777 strain), porcine circovirus type 2 (PCV2, SX07 strain), porcine reproductive and respiratory syndrome virus (PRRSV, TJM-F92 strain), foot-and-mouth disease virus (FMDV, O/Mya98/XJ/2010 strain), classical swine fever virus (CSFV, C strain), pseudorabies virus (PRV, Bartha-K61 strain), swine influenza virus (SIV, TJ strain), and porcine rotavirus (PoRV, NX strain).

The following positive samples were provided by our laboratory: PSV, PKV, PTV, EV-G, African swine fever virus (ASFV), and porcine deltacoronavirus (PDCoV).

### 2.2. Design of Primers and Probes

The genome sequences of PSV, PKV, PTV, and EV-G representative strains from different countries around the world were downloaded from GenBank in NCBI (https://www.ncbi.nlm.nih.gov/nucleotide/, accessed on 15 August 2023). The multiple sequence alignments were performed, and the conserved regions of the 5′ untranslated region (UTR) were selected for designing the specific primers and probes (Table 1), which are suitable for the detection of different strains of PSV, PKV, PTV, and EV-G from different countries. The genome accession number for the reference strains and the amplified gene fragments in the 5′ UTR of PSV, PKV, PTV, and EV-G are shown in Appendix A.

### 2.3. Clinical Samples

Between January 2024 and December 2024, a total of 1823 fecal samples from diarrheic piglets were collected in Guangxi Province in China. These samples came from diarrheal piglets less than 3 months old in different intensive pig farms in 14 cities of Guangxi Province during different seasons in 2024. Three-to-five grams of feces was collected from each diarrheal piglet using an anal swab. All samples were sent to the laboratory at ≤4 °C conditions within six hours post collection.

### 2.4. Extraction of Nucleic Acid

All the fecal samples were placed in 1.5 mL centrifuge tubes, and 1.0 mL of phosphate-buffered saline (PBS, pH7.2) (1:4, *w*/*v*) was added to each tube, vortexed (2 min), and centrifuged at 4 °C (12,000 rpm, 5 min). The nucleic acids of the clinical samples or vaccine solutions were extracted according to the MiniBEST Viral RNA/DNA Extraction Kit Ver.5.0 (TaKaRa, Dalian, China; Cat No. 9766) using 200 µL of supernatants or vaccine solutions. A 30 μL elution buffer was used to dissolve the extracted nucleic acids, which were used to generate standard plasmid constructs, or detect the target gene fragments.

### 2.5. Construction of Standard Plasmids

The synthesized viral RNAs of PSV, PKV, PTV, and EV-G corresponding to the target gene fragments for amplification were provided by Dalian TaKaRa Co., Ltd. (TaKaRa, Dalian, China). The RNAs were used for evaluating the sensitivity of the developed quadruplex RT-qPCR assay.

The standard plasmid constructs were generated according to the reported procedures [32] with minor modification. The extracted nucleic acids from PSV, PKV, PTV, and EV-G positive samples were reverse transcribed to cDNA using PrimeScript II 1st Strand cDNA Synthesis Kit (TaKaRa, Dalian, China; Cat No. 6210A). The corresponding target gene fragments were amplified from the cDNA using the specific primers in Table 1. The PCR products were purified using MiniBEST DNA Fragment Purification Kit Ver.4.0 (TaKaRa, Dalian, China; Cat No. 9762), ligated into pMD18-T vector (TaKaRa, Dalian, China; Cat No. 6011), and transformed into DH5α competent cells (TaKaRa, Dalian, China; Cat No. 9057). The positive clones were selected and incubated at 37 °C for 20–24 h, and then the plasmids were extracted using MiniBEST Plasmid Extraction Kit Ver.5.0 (TaKaRa, Dalian, China; Cat No. 9760). Finally, the recombinant standard plasmid constructs were named p-PSV, p-PKV, p-PTV, and p-EV-G. The supercoiled plasmids were treated with *Eco*RI endonuclease (TaKaRa, Dalian, China; Cat No. 1040A) to obtain linearized plasmids. Their concentrations were determined through a spectrophotometer at OD_260_/OD_280_ nm using the following equation: $$\mathrm{concentration}\;(\mathrm{copies}/\mu L)\;=\;\frac{\left(6.02\times 10^{23}\right)\times (X\;ng/µL\;\times 10^{-9})}{plasmid\;length\;(bp)\times 660}.$$

### 2.6. Optimization of Reaction System and Reaction Procedure

The reaction system for the quadruplex RT-qPCR assay was optimized using the ABI Q5 Real-Time System (Thermo Fisher scientific, Carlsbad, CA, USA). The primers and probes were diluted to 20 μM, and the total volume of the reaction system was 20 µL: 10 μL 2× One Step RT-PCR Buffer III (TaKaRa, Dalian, China; Cat No. RR064A), 0.4 µL Ex Taq HS (TaKaRa, Dalian, China; Cat No. RR064A), 0.4 µL PrimeScript RT Enzyme Mix II (TaKaRa, Dalian, China; Cat No. RR064A), 2.0 µL of a mixture of the four standard plasmid constructs (10^8^ copies/µL, and the final reaction concentration was 10^7^ copies/µL in the reaction system), forward/reverse primers and probes for four viruses 0.2–0.5 µL, and distilled water to a final volume of 20 µL. Fluorescence signals were collected at the end of each reaction cycle. The variables in this experiment were the concentrations of forward/reverse primers and probes, the temperatures of the reaction program, and reaction cycles to determine the optimal reaction system and reaction program. Their optimal conditions were based on the minimum cycling threshold (Ct) and the maximum ∆Rn at the end of the reaction.

### 2.7. Generation of Standard Curves

The plasmid constructs p-PSV, p-PKV, p-PTV, and p-EV-G were mixed at a ratio of 1:1:1:1, then 10-fold serially diluted. The mixtures with concentrations of 1.0 × 10^8^ to 1.0 × 10^2^ copies/µL (the final reaction concentration: 1.0 × 10^7^ to 1.0 × 10^1^ copies/µL) were used as templates to generate the standard curves of the developed quadruplex RT-qPCR assay.

### 2.8. Analytical Specificity Analysis

The total nucleic acids of PSV, PKV, PTV, EV-G, TGEV, PCV2, PRRSV, FMDV, CSFV, PRV, SIV, PoRV, ASFV, PEDV, and PDCoV were used as templates for specificity analysis of the developed quadruplex RT-qPCR assay. The plasmid constructs p-PSV, p-PKV, p-PTV, p-EV-G, the positive clinical samples, and nuclease-free distilled water were used as controls.

### 2.9. Analytical Sensitivity Analysis

The synthesized viral RNAs of PSV, PKV, PTV, and EV-G were mixed at a ratio of 1:1:1:1, and 10-fold serially diluted. The concentrations of the mixtures with 1.0 × 10^8^ to 1.0 × 10^1^ copies/µL (the final reaction concentration: 1.0 × 10^7^ to 1.0 × 10^0^ copies/µL) were used as templates for sensitivity analysis of the developed quadruplex RT-qPCR assay. The limits of detection (LODs) were determined.

In addition, the mixtures of synthesized viral RNAs of PSV, PKV, PTV, and EV-G with 500, 250, 125, and 62.5 copies/reaction were also used as templates for the sensitivity analysis of the developed assay using probit regression analysis.

### 2.10. Repeatability Analysis

The plasmid constructs p-PSV, p-PKV, p-PTV, and p-EV-G were mixed together at a ratio of 1:1:1:1, and 10-fold serially diluted. The plasmid construct concentrations of 1.0 × 10^7^, 1.0 × 10^4^, and 1.0 × 10^2^ copies/µL (the final concentrations: 1.0 × 10^6^, 1.0 × 10^3^, and 1.0 × 10^1^ copies/µL) were used as templates for the determination of the coefficient of variation (CV) of the developed quadruplex RT-qPCR assay. The intra-assay tests were performed in triplicates, and the inter-assay tests were performed on three different days.

### 2.11. Assessment of the Developed Assay Using Clinical Samples

The 1823 fecal samples collected from diarrheic pigs in Guangxi Province were tested using the established quadruplex RT-qPCR for evaluating its applicability. In addition, these samples were also tested using the RT-qPCR assays previously reported for the detection of PSV [33], PKV [34], and PTV [35], and using the RT-PCR assay previously reported for the detection of EV-G [29]. The diagnostic sensitivity and specificity of the developed quadruplex RT-qPCR assay were assessed. The detection results of the developed assay and the reference assays were compared, and the coincidence rates were calculated.

## 3. Results

### 3.1. Preparation of the Standard Plasmids

To construct the standard plasmids, total nucleic acids were extracted from the positive clinical samples of PSV, PKV, PTV, and EV-G, then reverse transcribed into cDNA. The amplified target fragments of the 5′ UTR of PSV, PKV, PTV, and EV-G using the primers in Table 1 were purified and used for the construction of recombinant standard plasmid constructs. The obtained plasmid constructs were named p-PSV, p-PKV, p-PTV, and p-EV-G, and their concentrations were measured by a spectrophotometer. The initial concentrations of p-PSV, p-PKV, p-PTV, and p-EV-G were calculated as 4.83 × 10^10^, 5.23 × 10^10^, 5.45 × 10^10^, and 5.17 × 10^10^ copies/µL, respectively. Finally, they were diluted to 1.0 × 10^10^ copies/µL, and stored at −80 °C until use.

### 3.2. Attainment of the Optimal Reaction Parameters

To develope the quadruplex RT-qPCR, the p-PSV, p-PKV, p-PTV, and p-EV-G plasmid constructs were used as templates, and the One-Step PrimeScript RT-PCR Kit (TaKaRa, Dalian, China; Cat No. RR064A) was used as basic components. The reaction conditions of the quadruplex RT-qPCR assay were optimized, including primer and probe concentrations, annealing temperatures, and reaction cycles. After optimization, the reaction system and amplification procedures for the quadruplex RT-qPCR were obtained. The ingredients of the 20 µL reaction system is shown in Table 2. The optimized reaction protocol was as follows: 42 °C for 5 min, 95 °C for 10 s; followed by 40 cycles of 95 °C for 5 s and 56 °C for 30 s. The fluorescence signals were collected after each cycle, and the samples with Ct values ≤36 were judged as positive samples.

### 3.3. Generation of the Standard Curves

The four plasmid constructs were mixed in equal proportions and serially diluted. The mixtures with concentrations of 1.0 × 10^8^–1.0 × 10^2^ copies/µL (the final reaction concentration: 1.0 × 10^7^–1.0 × 10^1^ copies/µL) were used to generate the standard curves of the developed assay (Figure 1). The results indicated that the amplification efficiencies (Eff%) of the standard curves for PSV, PKV, PTV, and EV-G were 105.45%, 102.871%, 101.102%, and 98.776%, respectively; their correlation coefficients (R^2^) were 1, 0.999, 1, and 0.999, respectively, indicating excellent amplification efficiency and correlation coefficient (R^2^ ≥ 0.998) of the quadruplex RT-qPCR.

### 3.4. Specificity of the Quadruplex RT-qPCR

The specificity of the established assay was assessed using the total nucleic acids of PSV, PKV, PTV, EV-G, TGEV, PCV2, PRRSV, FMDV, CSFV, PRV, SIV, PoRV, ASFV, PEDV, and PDCoV. The results showed that the quadruplex RT-qPCR produced specific amplification curves only for PSV, PKV, PTV, and EV-G, but not for the other control viruses (Figure 2), indicating the strong specificity of the quadruplex RT-qPCR.

### 3.5. Sensitivity of the Quadruplex RT-qPCR

The sensitivity of the developed quadruplex RT-qPCR was evaluated using the synthesized viral RNAs. The mixtures of synthesized viral RNAs of PSV, PKV, PTV, and EV-G with concentrations from 1.0 × 10^8^ to 1.0 × 10^1^ copies/µL (the final concentration: 1.0 × 10^7^ to 1.0 × 10^0^ copies/µL). The results indicated that the LODs were 1.0 × 10^1^ copies/µL for all four synthesized viral RNAs (Figure 3).

The LODs were also assessed using probit regression analysis. The concentrations of 500, 250, 125, and 62.5 copies/reaction for the synthesized viral RNAs of PSV, PKV, PTV, and EV-G were selected as templates (Table 3). The results showed that PSV, PKV, PTV, and EV-G had LODs of 146.02 (95% confidence interval (CI) of 134.34–171.12), 143.83 (95% CI of 132.58–166.52), 141.92 (95% CI of 130.95–162.92), and 139.79 (95% CI of 129.07–159.29), respectively (Figure 4), indicating high sensitivity of the quadruplex RT-qPCR.

### 3.6. Repeatability of the Quadruplex RT-qPCR

The repeatability of the developed quadruplex RT-qPCR was analyzed using the mixtures of plasmid constructs p-PSV, p-PKV, p-PTV, and p-EV-G, with concentrations of 1.0 × 10^7^, 1.0 × 10^4^, and 1.0 × 10^2^ copies/µL (the final concentrations: 1.0 × 10^6^, 1.0 × 10^3^, and 1.0 × 10^1^ copies/µL), respectively. The results indicated that the intra-assay CVs were 0.27–1.57%, and the inter-assay CVs were 0.21–1.41% (Table 4), indicating the excellent reproducibility of the developed assay.

### 3.7. Assessment Results of the Clinical Samples

The 1823 fecal samples collected from diarrheic piglets in Guangxi Province between January 2024 and December 2024 were tested using the developed quadruplex RT-qPCR assay. The results indicated that the positivity rates of PSV, PKV, PTV, and EV-G were 15.25% (278/1823), 21.72% (396/1823), 18.82% (343/1823), and 27.10% (494/1823), respectively. Furthermore, mixed infections of these viruses were also found in the clinical samples (Table 5). PTV + EV-G co-infections showed the highest positivity rate of dual infection at 13.76% (251/1823), and quadruple co-infections were detected in 1.91% (35/1823) of these fecal samples.

In addition, the reference RT-qPCR for the detection of PSV [33], PKV [34], and PTV [35], and the reference RT-PCR for the detection of EV-G [29] were used to test these 1823 fecal samples. The results showed that the positivity rates of PSV, PKV, PTV, and EV-G were 15.03% (274/1823), 21.56% (393/1823), 18.65% (340/1823), and 26.82% (488/1823), respectively. Compared with the reported reference methods, the developed quadruplex RT-qPCR assay showed diagnostic sensitivity and specificity for PSV, PKV, PTV, and EV-G at 99.27% and 99.61%, 98.98% and 99.51%, 99.12% and 99.60%, and 98.77% and 99.10%, respectively (Table 6). The two methods showed coincidence rates of 99.56%, 99.40%, 99.51%, and 99.01% for PSV, PKV, PTV, and EV-G, respectively (Appendix A).

## 4. Discussion

At present, many viruses can cause diarrhea in pig herds [36,37,38]; of which, PSV, PKV, PTV, and EV-G are common viruses that play important roles in the establishment of diarrhea in animals. PSV was initially discovered in the United Kingdom in 1958, and the first Chinese PSV strain was isolated in Shanghai Municipality in 2009 [39]. There is a widespread prevalence of PSV in porcine populations, and even a positivity rate as high as 79.3% in some areas [5,9,40]. PKV has been reported globally since its discovery in 2007 [41,42]. Even worse, the recombinant strains of PKV were found in the circulating strains [43,44]. The first outbreak of PTV occurred in Czechoslovakia in 1929, and different genotypes have been found. PTV1 has a high prevalence in wild and domestic pigs in some European countries [45]. In China, PTV2 infection was first identified in Heilongjiang Province in 2009 [46], and has also been found in Guangxi Province [47]. Recently, six strains of known genotypes, and four undefined genotype strains were obtained in Jiangxi Province, indicating there were more than two genotypes of PTV2 in China [48]. This indicated that PTV had genetic diversity in China. EV-G was first isolated in 1973. Diarrheic pigs infected with EV-G have been reported in many countries [28,49,50]. EV-G was also widespread in Guangxi Province, and had a high positivity rate in pig farms, even reaching a 100% positivity rate [51]. The prevalence of EV-G (24.37%, 97/398) was higher than those of PSV (5.77%, 23/398) and PTV (12.81%, 51/398), and there also existed co-infections of these three viruses in India [52]. These situations make it necessary and urgent to establish accurate detection methods, and perform investigation and epidemiology studies on these pathogens.

PSV, PKV, PTV, and EV-G play important roles in the establishment of diarrhea in pig herds, and they show similar clinical signs and pathological changes, which are hard to differentiate in clinical cases. Even worse, co-infections with these viruses are usually found [52,53], and these exacerbate the diarrhea and other clinical signs [14,28,52,54]. Current techniques used for the detection and identification of viruses usually include electron microscopy, virus isolation, and serological tests. However, they show the shortcomings of cumbersomeness, relatively low sensitivity, and susceptibility to environmental contamination. On the contrary, qPCR can detect the target nucleic acids based on Ct values. It has the advantages of easy operation, strong specificity, and high sensitivity, and has been widely used in different biological laboratories [30,31]. To date, there exist single RT-qPCR assays for the detection of PSV, PKV, and PTV [33,34,35], and a multiplex RT-PCR assay for the detection of PSV, PTV, and EV-G [29]. However, no multiplex RT-qPCR assay has been reported for the simultaneous detection of these four viruses until now. In this study, the standard plasmid constructs were prepared, the reaction conditions were optimized, the standard curves were generated, and the specificity, sensitivity, and reproducibility were evaluated. The results indicated that the developed quadruplex RT-qPCR assay specifically detected PSV, PKV, PTV, and EV-G; it showed low LODs of 146.02, 143.83, 141.92, and 139.79 copies/reaction for PSV, PKV, PTV, and EV-G, respectively, which had higher sensitivity than the reference RT-qPCR for PSV (934 copies/reaction), PKV (750 copies/reaction), PTV (200 copies/reaction) [33,34,35], and the non-quantitative analysis of EV-G due to RT-PCR [29]. The assay indicated intra-assay CVs of 0.27–1.57% and inter-assay CVs of 0.21–1.41%. Furthermore, 1823 fecal samples collected in Guangxi Province were validated using the established quadruplex RT-qPCR and the previously reported reference RT-qPCR/RT-PCR [29,33,34,35]. The detection results showed that their coincidence rates were higher than 99.01%. These results indicated that the quadruplex RT-qPCR assay had the advantages of easy operation, high specificity, excellent sensitivity, and good reproducibility, and was suitable for application to detect clinical samples. It is noteworthy that there exists genetic diversity in the prevalent strains of PSV, PKV, PTV, and EV-G, so the design of the specific primers and probes is very important for the development of a quadruplex RT-qPCR. In this study, the multiple sequence alignments of viral strains from different countries were performed. Then, the conserved regions in 5′ UTR of PSV, PKV, PTV, and EV-G were selected as the target fragments to design the specific primers and probes, which ensured the detection of different viral strains prevalent in various countries around the world. Of course, due to the continuous variation of the epidemic strains, it is necessary to continuously perform molecular epidemiological research. This ensures the grasping of the viral genetic variation and the adjustment of primers and probe sequences on time, and ensures the accurate detection of the epidemic strains.

In this study, 1823 fecal samples collected from different pig farms in Guangxi Province had positivity rates of 15.03%, 21.56%, 18.65%, and 26.82% for PSV, PKV, PTV, and EV-G, respectively. This suggested that these viruses have high positivity rates in Guangxi Province. These viruses have also been reported in many provinces of China [5,6,8,9,13,14,16,17,18,20,23,24,25], and many countries around the world [2,3,4,7,18,21,22,27,36,44,50,55,56]. In China, it has been found that PKV is usually co-infected with PEDV, and PKV may enhance the virulence of PEDV, resulting in more serious damage to the hosts [13,14]. In our previous study, the positivity rate of PEDV during 2020–2024 in Guangxi Province was 11.90% (397/5859) [57]. In this study, the 1823 fecal samples were also tested for PEDV using the RT-qPCR assay previously established in our laboratory [37]. The results showed that PEDV had a positivity rate of 5.38% (98/1823); PKV and PEDV had a co-infection rate of 3.46% (63/1823), while PKV (+)/PEDV (−) was 18.10% (330/1823), and PEDV (+)/PKV (−) was 1.92% (35/1823). The results showed that PKV and PEDV co-infection existed in Guangxi Province, and further research is needed to evaluate the synergism of PKV and PEDV. In this study, triple and quadruple co-infections of PSV, PKV, PTV, and EV-G were also detected in fecal samples, with the highest positivity rate of 13.76% (251/1823) for co-infections of PTV and EV-G. In the Czech Republic, co-infections of PTV and EV-G were also as high as 30.4% (17/56) in wild boars and 12.9% (16/124) in domestic pigs [28]. PSV, PTV, and EV-G co-infections have also been reported in India and Brazil [52,53]. At present, all four viruses are widely spread in different countries around the world, and co-infections of pigs with *Picornaviridae* viruses are common, and the harms of these viruses cannot be ignored. It is urgent to carry out appropriate prevention and control measures to curb the spread of these viruses and decrease the economic losses of these diseases.

Especially, PSV [6,7] and PKV [11,12,58,59] have the zoonotic potentials to infect humans. The widespread occurrence of PSV and PKV creates a risk of cross-species transmission, and induces huge harms to the pig industry and human health. This further highlights the urgency and importance of developing the quadruplex RT-qPCR for the rapid, accurate, and simultaneous detection of PSV, PKV, PTV, and EV-G. In addition, due to the zoonotic potentials of PSV and PKV, there is indeed a risk of these viruses being transmitted from pigs across species to humans. Therefore, it is extremely important to take strong prevention and control measures to prevent these viruses from being transmitted from pigs to humans. The measures include but are not limited to the following main measures: establish and implement strict biosecurity measures in pig farms; regularly clean and disinfect the external environment thoroughly; use personal protective measures for workers and veterinarians who have close contact with pigs; clean and disinfect vehicles and items before entering the farms; timely dispose of sick pigs to reduce their chances of contact with humans.

## 5. Conclusions

The developed quadruplex RT-qPCR assay for the detection of PSV, PKV, PTV, and EV-G is characterized by easy operation, high sensitivity, and strong specificity, and can be used for the accurate evaluation of PSV, PKV, PTV, and EV-G in clinical specimens. The detection results of clinical samples indicated that these viruses are widespread in Guangxi Province, southern China, and further experimental studies demonstrating their roles in the establishment of diarrhea in pigs would be interesting.

## Figures and Tables

**Figure 1 animals-15-01008-f001:**
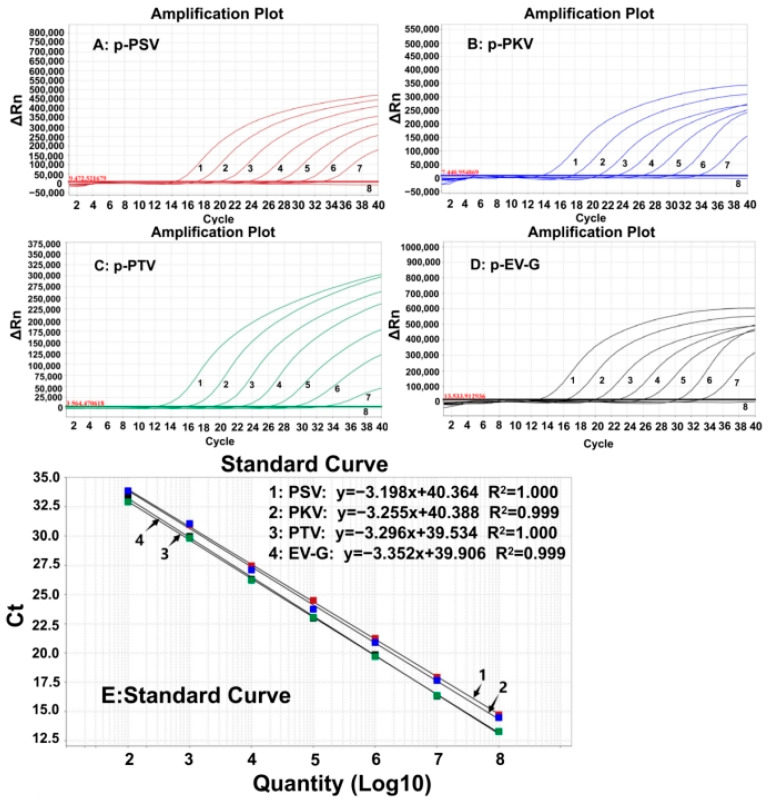
The amplification curves of p-PSV (**A**), p-PKV (**B**), p-PTV (**C**), and p-EV-G (**D**), and the standard curves (**E**) of the developed quadruplex RT-qPCR. In subfigure (**A**–**D**) 1–7, the final concentration of the plasmid constructs ranged from 10^7^ to 10^1^ copies/µL; 8: distilled water.

**Figure 2 animals-15-01008-f002:**
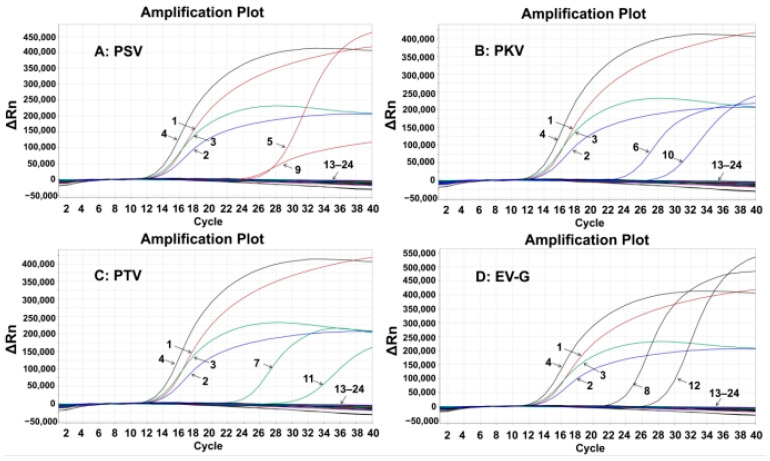
Specificity evaluation of the developed quadruplex RT-qPCR for PSV (**A**), PKV (**B**), PTV (**C**), and EV-G (**D**). 1: p-PSV; 2: p-PKV; 3: p-PTV; 4: p-EV-G; 5–8: positive clinical samples of PSV, PKV, PTV, and EV-G; 9: PSV; 10: PKV; 11: PTV; 12: EV-G; 13–22: TGEV, PCV2, PRRSV, FMDV, CSFV, PRV, SIV, PoRV, ASFV, PEDV, and PDCoV; 23: negative fecal sample; 24: distilled water.

**Figure 3 animals-15-01008-f003:**
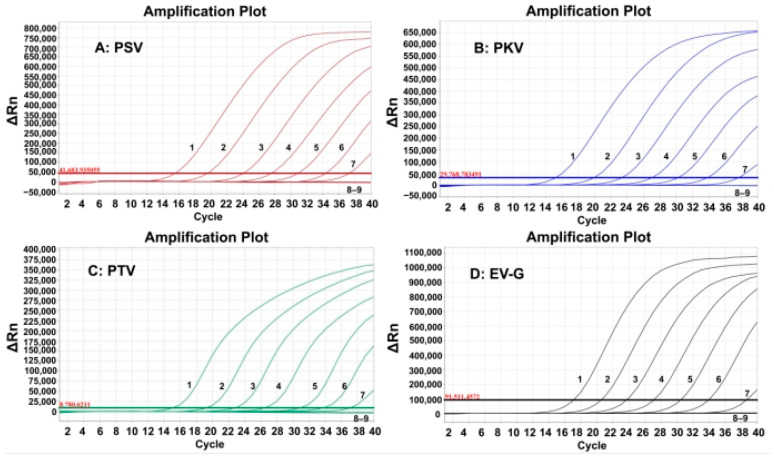
Sensitivity evaluation of the developed quadruplex RT-qPCR. In subfigures (**A**–**D**) 1–8: the final reaction concentrations of the synthesized viral RNAs ranged from 10^7^ to 10^0^ copies/µL; 9: negative control.

**Figure 4 animals-15-01008-f004:**
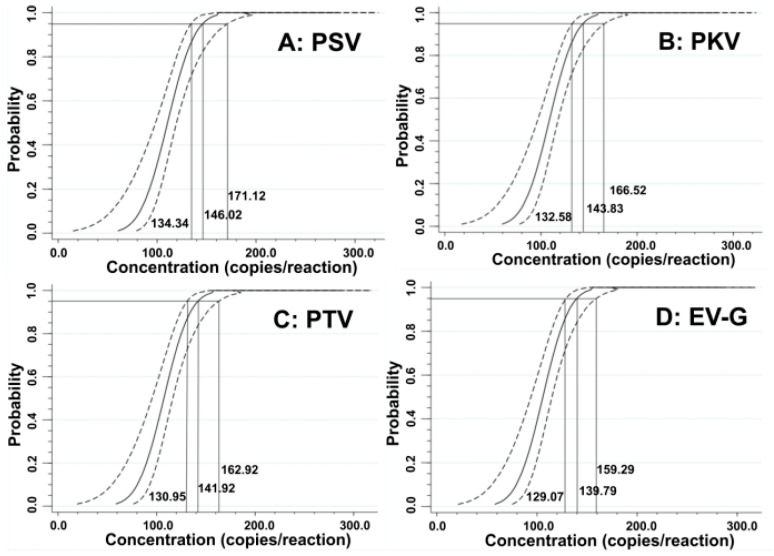
Sensitivity evaluation of the developed quadruplex RT-qPCR using probit regression analysis. The PSV (**A**), PKV (**B**), PTV (**C**), and EV-G (**D**) had LODs of 146.02, 143.83, 141.92, and 139.79 copies/reaction, respectively.

**Table 1 animals-15-01008-t001:** The designed specific primers and probes.

Primer/Probe	Sequence (5′ → 3′)	Gene	Tm/°C	Product (bp)
PSV-F	GACTGGGCCTATACTACCTGATA	5′ UTR	57.2	140
PSV-R	AGGTACACACGGGCTCTCTG		60.6	
PSV-P	ROX-TGGCCGCCTGTAACTAGTATAGTCAGT-BHQ2		63.6	
PKV-F	CGTGCTGAGTAATGGGATAGG	5′ UTR	57.0	149
PKV-R	TGCACTTCAGAGGTCAGAGAA		57.9	
PKV-P	VIC-ATGAGTAGAGCATGGACTGCGGTG-BHQ1		63.4	
PTV-F	GGACTGCRTTGCATATCCCTA	5′ UTR	58.3	137
PTV-R	GACTATACAAAGTACAGACGGCCA		58.9	
PTV-P	CY5-CTGTATGGGAATGCAGGACTGG-BHQ3		59.9	
EV-G-F	GGACCTAGTAGTGATAGGCTGTA	5′ UTR	57.1	113
EV-G-R	CTGAGCGAAACGCCAAGA		57.2	
EV-G-P	FAM-GCCGAAGATGAACCCGTCCGTTAT-BHQ1		63.8	

Note: In degenerate primers, R represents A or G. F—forward; R—reverse; P—probe. The same as follows.

**Table 2 animals-15-01008-t002:** The components and their optimal parameters.

Ingredient	Volume (µL)	Final Concentration (nM)
2× One-Step RT-PCR Buffer III	10.0	/
Ex Taq HS (5 U/µL)	0.4	/
PrimeScript RT Enzyme Mix II	0.4	/
PSV-F	0.3	300
PSV-R	0.3	200
PSV-P	0.2	200
PKV-F	0.2	200
PKV-R	0.2	200
PKV-P	0.2	200
PTV-F	0.2	200
PTV-R	0.2	200
PTV-P	0.2	200
EV-G-F	0.2	200
EV-G-R	0.2	200
EV-G- P	0.2	200
Total Nucleic Acid	2.0	/
Nuclease-Free Distilled Water	Up to 20.0	/

**Table 3 animals-15-01008-t003:** The Ct values and hit rates of the serially diluted synthesized viral RNAs.

RNA	Copies/Reaction	Number of Samples	Quadruplex RT-qPCR	
			Ct Value ($\bar{X}$ ± SD)	Hit Rate (%)
PSV	500	36	34.93 ± 0.13	100
	250	36	35.49 ± 0.12	100
	125	36	35.89 ± 0.13	75.0
	62.5	36	ND	0
PKV	500	36	34.86 ± 0.11	100
	250	36	35.41 ± 0.13	100
	125	36	35.83 ± 0.15	77.8
	62.5	36	ND	0
PTV	500	36	34.73 ± 0.10	100
	250	36	35.37 ± 0.13	100
	125	36	35.76 ± 0.15	80.6
	62.5	36	ND	0
EV-G	500	36	34.67 ± 0.14	100
	250	36	35.33 ± 0.10	100
	125	36	34.73 ± 0.13	83.3
	62.5	36	ND	0

Note: ND—not detected.

**Table 4 animals-15-01008-t004:** Repeatability of the developed quadruplex RT-qPCR.

Plasmid	Concentration (Copies/μL)	Ct Values of Intra-Assay			Ct Values of Inter-Assay		
		$\bar{X}$	SD	CV (%)	$\bar{X}$	SD	CV (%)
p-PSV	1.0 × 10^6^	15.92	0.25	1.57%	15.45	0.12	0.78%
	1.0 × 10^3^	25.89	0.08	0.31%	25.88	0.10	0.39%
	1.0 × 10^1^	33.94	0.15	0.44%	33.56	0.11	0.33%
p-PKV	1.0 × 10^6^	15.85	0.19	1.20%	15.26	0.17	1.11%
	1.0 × 10^3^	26.17	0.13	0.50%	26.04	0.08	0.31%
	1.0 × 10^1^	33.65	0.25	0.74%	33.83	0.07	0.21%
p-PTV	1.0 × 10^6^	15.43	0.15	0.97%	15.58	0.07	0.45%
	1.0 × 10^3^	26.59	0.09	0.34%	26.52	0.09	0.34%
	1.0 × 10^1^	33.11	0.09	0.27%	33.02	0.11	0.33%
p-EV-G	1.0 × 10^6^	15.45	0.23	1.49%	14.93	0.21	1.41%
	1.0 × 10^3^	25.57	0.08	0.31%	25.36	0.13	0.51%
	1.0 × 10^1^	32.39	0.12	0.37%	32.37	0.30	0.93%

**Table 5 animals-15-01008-t005:** Assessment results of the clinical specimens from different cities in Guangxi Province.

Region	Number	Positive Sample														
		PSV	PKV	PTV	EV-G	S + K	S + T	S + E	K + T	K + E	T + E	S + K + T	S + T + E	S + K + E	K + T + E	S + K + T + E
Nanning	197	7	76	15	22	1	4	3	12	18	10	1	2	1	9	1
Liuzhou	240	9	23	33	14	6	4	4	9	10	6	4	3	3	5	3
Guilin	23	0	1	1	1	0	0	0	0	0	0	0	0	0	0	0
Wuzhou	54	0	0	0	0	0	0	0	0	0	0	0	0	0	0	0
Beihai	20	0	0	0	0	0	0	0	0	0	0	0	0	0	0	0
Fangchenggang	88	6	24	8	22	0	1	1	0	6	2	1	4	1	1	1
Qinzhou	50	0	0	0	0	0	0	0	0	0	0	0	0	0	0	0
Guigang	221	68	125	64	136	31	40	66	30	88	58	11	40	31	27	11
Yulin	124	10	27	21	18	4	2	2	3	10	6	0	0	2	3	0
Baise	482	137	77	158	202	19	64	101	73	68	134	19	61	19	68	19
Hezhou	93	38	12	29	57	1	20	35	1	6	27	0	20	0	1	0
Hechi	39	2	0	5	2	0	1	1	0	0	2	0	1	0	0	0
Laibin	100	0	0	0	0	0	0	0	0	0	0	0	0	0	0	0
Chongzuo	92	1	31	9	20	0	0	0	9	20	6	0	0	0	6	0
Total	1823	278 (15.25%)	396 (21.72%)	343 (18.82%)	494 (27.10%)	62 (3.40%)	136 (7.46%)	213 (11.68%)	137 (7.51%)	226 (12.39%)	251 (13.76%)	36 (1.97%)	131 (7.18%)	57 (3.12%)	120 (6.58%)	35 (1.91%)

Note: S + K denotes co-infection of PSV and PKV; S + T denotes co-infection of PSV and PTV; S + E denotes co-infection of PSV and EV-G; K + T denotes co-infection of PKV and PTV; K + E denotes co-infection of PKV and EV-G; T + E denotes co-infection of PTV and EV-G; S + K + T denotes co-infection of PSV, PKV, and PTV; S + T + E denotes co-infection of PSV, PTV, and EV-G; S + K + E denotes co-infection of PSV, PKV, and EV-G; K + T + E denotes co-infection of PKV, PTV, and EV-G; S + K + T + E denotes co-infection of PSV, PKV, PTV, and EV-G. In this study, the Ct values of the positive samples were as follows (Ct ± SD): PSV: 24.54 ± 4.91; PKV: 28.31 ± 3.57; PTV: 28.16 ± 4.05; EV-G: 26.10 ± 5.11.

**Table 6 animals-15-01008-t006:** The diagnostic sensitivity and specificity of the developed assay.

The Developed Assay		The Reference Assay		Total	Diagnostic Sensitivity (95% CI)	Diagnostic Specificity (95% CI)
		Positive	Negative			
PSV	Positive	272	6	278	99.27% (97.38–99.80%)	99.61% (99.16–99.82%)
	Negative	2	1543	1545		
	Total	274	1549	1823		
PKV	Positive	389	7	396	98.98% (97.41–99.60%)	99.51% (98.99–99.76%)
	Negative	4	1423	1427		
	Total	393	1430	1823		
PTV	Positive	337	6	343	99.12% (97.44–99.70%)	99.60% (99.12–99.81%)
	Negative	3	1477	1480		
	Total	340	1483	1823		
EV-G	Positive	482	12	494	98.77% (97.34–99.44%)	99.10% (98.44–99.49%)
	Negative	6	1323	1329		
	Total	488	1335	1823		

## Data Availability

The original contributions presented in this study are included in this article/Appendix A; further inquiries can be directed to the corresponding authors.

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
