# Peer review of "A Quadruplex RT-qPCR for the Detection of Porcine Sapelovirus, Porcine Kobuvirus, Porcine Teschovirus, and Porcine Enterovirus G"

_animals, 2025, doi:10.3390/ani15071008_

Round 1
Reviewer 1 Report
Comments and Suggestions for Authors
The authors developed a quadruplex RT-qPCR for the simultaneous detection of porcine Sapelovirus, Kobuvirus, Teschovirus and Enterovirus G. They also assessed the sensitivity, specificity and repeatability of their technique with reference techniques for detecting these agents individually.
The article is well written, the methodology is consistent, and the manuscript describes the first quadruplex RT-qPCR for detection of the above viruses simultaneously.
There are some concepts from the introduction that should be modified, as well as from the discussion and conclusions. With some minor corrections, I believe the manuscript could be of enough quality for publication.
Minor concerns
1) Lines 13-14: To our knowledge, these four viruses have not been shown to cause diarrhoea by themselves in swine. They have been associated with outbreaks of diarrhoea, but in most cases they have always been linked to other well-recognized agents of diarrhoea (E.coli, Rotavirus, PEDV, Salmonella, etc.). This statement is somewhat risky and is mentioned several times throughout the manuscript. They should modify it by mentioning that they could play a role in the establishment of diarrhoea in swine.
2) Line 24-25: The same as in the above point with respect to diarrhoea.
3) Lines 88-89: The same problem, we do not know if by themselves they can cause diarrhoea in pigs. Modify the assertion.
4) Lines 155-156: Perhaps it would be more appropriate to write that the primers have been diluted to a concentration of 20 µM, as this would help the reader to quickly understand the units.
5) Lines 205-210: This paragraph is redundant, it has already been mentioned in material and methods.
6) Table 2: Modify the volume of nucleic acid you add to the PCR reaction (I think to 2 µL), 20 µL is not possible.
7) Table 5: Try to keep the ‘Total’ row (the last row) to a maximum of 2 lines, as is the case for the column S+K+T+E.
8) Lines 311 and 331: They do not seem “important” agents that produce diarrhoea in animals (and even less so in pigs), there are other infectious agents that are much more important.
9) Lines 343-347: Again, the information in these lines is redundant. Adapt it or delete it.
10) Lines 357-361: I would not include these lines in the discussion, but rather place them where you talk about the design of primers and probes in material and methods.
11) Conclusions (Lines 393-397): I would add a concluding paragraph mentioning that since we have observed that these agents are prevalent in swine farms, experimental studies demonstrating their role in the establishment of diarrhoea in pigs would be interesting.
Recommendations
1) Line 69: As in the case of PKoV with PEDV, would there be further synergy studies with other well-recognised enteric agents for the other virus?
2) Table 7: It could be deleted, the information given in the text would be sufficient. It could also be sent to supplementary.
Author Response
The Cover Letter
March 6, 2025
Dear editor,
Our manuscript has been revised carefully according to the reviewers' suggestions. Please see the details as follows.
Reviewer #1
Comments and Suggestions for Authors
The authors developed a quadruplex RT-qPCR for the simultaneous detection of porcine Sapelovirus, Kobuvirus, Teschovirus and Enterovirus G. They also assessed the sensitivity, specificity and repeatability of their technique with reference techniques for detecting these agents individually.
The article is well written, the methodology is consistent, and the manuscript describes the first quadruplex RT-qPCR for detection of the above viruses simultaneously.
There are some concepts from the introduction that should be modified, as well as from the discussion and conclusions. With some minor corrections, I believe the manuscript could be of enough quality for publication.
Minor concerns
1) Lines 13-14: To our knowledge, these four viruses have not been shown to cause diarrhoea by themselves in swine. They have been associated with outbreaks of diarrhoea, but in most cases they have always been linked to other well-recognized agents of diarrhoea (E. coli, Rotavirus, PEDV, Salmonella, etc.). This statement is somewhat risky and is mentioned several times throughout the manuscript. They should modify it by mentioning that they could play a role in the establishment of diarrhoea in swine.
Response: We agree to the reviewer's suggestion. These sentences have been revised. Please see Lines 13-15, Lines 25-27, Lines 91-92, Lines 330-332, and Lines 352-354 in the revised manuscript.
2) Line 24-25: The same as in the above point with respect to diarrhoea.
Response: We agree to the reviewer's suggestion. The sentence has been revised. Please see Lines 25-27 in the revised manuscript.
3) Lines 88-89: The same problem, we do not know if by themselves they can cause diarrhoea in pigs. Modify the assertion.
Response: We agree to the reviewer's suggestion. The sentence has been revised. Please see Lines 91-92 in the revised manuscript.
4) Lines 155-156: Perhaps it would be more appropriate to write that the primers have been diluted to a concentration of 20 µM, as this would help the reader to quickly understand the units.
Response: We agree to the reviewer's suggestion. “20 μL/pmol” has been changed to “20 μM”. Please see Lines 168-169 in the revised manuscript.
5) Lines 205-210: This paragraph is redundant, it has already been mentioned in material and methods.
Response: We agree to the reviewer's suggestion. These sentences have been revised to briefly describe the process of generation of the standard plasmids. Please see Lines 219-225 in the revised manuscript.
6) Table 2: Modify the volume of nucleic acid you add to the PCR reaction (I think to 2 µL), 20 µL is not possible.
Response: We agree to the reviewer's suggestion. “20” has been changed to “2.0”. Please see Table 2 in the revised manuscript.
7) Table 5: Try to keep the ‘Total’ row (the last row) to a maximum of 2 lines, as is the case for the column S+K+T+E.
Response: We agree to the reviewer's suggestion. The last row has been revised to 2 lines. Please see Table 5 in the revised manuscript.
8) Lines 311 and 331: They do not seem “important” agents that produce diarrhoea in animals (and even less so in pigs), there are other infectious agents that are much more important.
Response: We agree to the reviewer's suggestion. “important” has been changed to “common”. Please see Line 331, and Line 351 in the revised manuscript.
9) Lines 343-347: Again, the information in these lines is redundant. Adapt it or delete it.
Response: We agree to the reviewer's suggestion. The sentence has been revised. Please see Lines 364-369 in the revised manuscript.
10) Lines 357-361: I would not include these lines in the discussion, but rather place them where you talk about the design of primers and probes in material and methods.
Response: We agree to the reviewer's suggestion. The sentence has been deleted, and the corresponding content has been moved to Section 2.2. Design of Primers and Probes. Please see Lines 122-131, and 381-386 in the revised manuscript.
11) Conclusions (Lines 393-397): I would add a concluding paragraph mentioning that since we have observed that these agents are prevalent in swine farms, experimental studies demonstrating their role in the establishment of diarrhoea in pigs would be interesting.
Response: We agree to the reviewer's suggestion. The content has been added. Please see Lines 420-422 in the revised manuscript.
Recommendations
1) Line 69: As in the case of PKoV with PEDV, would there be further synergy studies with other well-recognised enteric agents for the other virus?
Response: We have not seen other papers until now. Thanks.
2) Table 7: It could be deleted, the information given in the text would be sufficient. It could also be sent to supplementary.
Response: We agree to the reviewer's suggestion. Table 7 has been moved to Supplementary Table S1. Please see Table S1 in the revised manuscript.
Reviewer #2
Comments and Suggestions for Authors
In this study, Li et al. developed a quadruplex RT-qPCR assay for the simultaneous detection of PSV, PKV, PTV, and EV-G. This assay offers a rapid, sensitive, and accurate method for detecting all four viruses concurrently. Overall, the experimental design is reasonable and the results are reliable. However, there are some issues that need to be addressed.
- The genome accession number for the reference strain must be provided.
Response: We agree to the reviewer's suggestion. The genome accession numbers for the reference strains have been provided in the Supplementary Figure S1. Please see Figure S1 in the revised manuscript.
- The detailed description of the sample information is lacking.
Response: We agree to the reviewer's suggestion. The detailed description of the sample information has been added. Please see Lines 134-140 in the revised manuscript.
- Please ensure that the subtitles of sections 2.8 and 2.9 are consistent; verify and make any necessary modifications.
Response: We agree to the reviewer's suggestion. The subtitles of sections 2.8 and 2.9 are consistent. Please see Lines 186 and 192 in the revised manuscript.
- The sensitivity of the detection method needs to be reflected in the abstract.
Response: We agree to the reviewer's suggestion. The information on the sensitivity of the detection method has been added. Please see Lines 33-35 in the revised manuscript.
Reviewer #3
Comments and Suggestions for Authors
The authors developed an RT-qPCR for the detection of four picornaviruses of interest in the swine industry. These viruses are primarily associated with gastrointestinal signs, such as diarrhea.
Following the design stage, recombinant pMD18-T vectors carrying each of the viral specific fragments were constructed, and the standard curves were built.
The analysis of the sensitivity and repeatability was carried out using, in addition to the four previously mentioned viruses, the genomes of other common swine viruses, including TGEV, PCV2, PRRSV, FMDV, CSFV, PRV, SIV, PoRV, ASFV, PEDV, and PDCoV.
The evaluation of the RT-qPCR technique was performed with 1,823 clinical samples obtained from Guangxi province in China. In the fine-tuning process with these clinical samples, the authors demonstrated the specificity of the designed primers, as no amplification of the genome of other common swine viruses was observed. Additionally, the study demonstrated the RT-qPCR technique's high sensitivity, with an average detection of 138 copies per microliter, accompanied by satisfactory inter- and intra-assay repeatability values.
The clinical samples were analyzed, yielding positive percentages of 15.25% (PSV), 21.72% (PKV), 18.82% (PTV), and 27.10% (EV-G). The presence of multiple viruses in various regions was determined, indicating the circulation of the viruses under study.
The study is methodically structured and written, meeting the minimum criteria for acceptance by the journal.
Minor comments:
The paper highlights the presence of principally PTV genotype 2 in China, as well as other genotypes not yet identified. The question of whether this was taken into account in the primer design was resolved, as the sequence alignment showed the name of at least 11 genotypes.
Response: We agree to the reviewer's suggestion. The genome sequences of PSV, PKV, PTV, and EV-G representative strains from different countries around the world were downloaded from GenBank in NCBI. The multiple sequence alignments were performed, and the conserved regions of the 5' untranslated region (UTR) were selected for designing the specific primers and probes, which are suitable for the detection of different strains of PSV, PKV, PTV and EV-G from different countries. Please see Lines 122-131 in the revised manuscript.
Reviewer #4
Comments and Suggestions for Authors
The study presented by Biao Li and collaborators describes the establishment of a quadruplex RT-qPCR for the molecular diagnosis of several Picornaviruses of interest in swine farming. The methodology used is adequately described and the findings of the study are correctly discussed.
- Lane 63 and others, change the word asymptomatic to subclinical.
Response: We agree to the reviewer's suggestion. “asymptomatic” has been changed to “subclinical”. Please see Lines 67, 74, and 93 in the revised manuscript.
- Lane 72 and others, change the word symptoms to signs.
Response: We agree to the review's suggestion. “symptoms” has been changed to “signs”. Please see Lines 75, 86, and 94 in the revised manuscript.
M&M
- Lane 107, it is certainly very useful to use reference strains of related viruses that infect pigs, some even from different families or genera. What about the other viruses of the Piconaviridae family that infect pigs? Why didn't you consider it? Did you perform in silico analysis of your primers and probes that include members of this viral family?
Response: We agree to the reviewer's suggestion. In our study, beside PSV, PKV, PTV, and EV-G, FMDV as the other member of the Piconaviridae family was used to evaluate the specificity of the developed RT-qPCR. To design the specific primers and probe, we also analyzed the genome sequences of other viruses in the Piconaviridae family. Please see Lines 113-119 in the revised manuscript.
- Lane 126, indicate the age range of the piglets from which the samples were obtained. Mention whether the samples came from intensive production farms or from some other type.
Response: We agree to the reviewer's suggestion. The samples came from intensive production farms. The information has been added. Please see Lines 134-140 in the revised manuscript.
- Lane 134, the manufacturer does not have feces on its list of recommended samples, how did you verify this?
Response: We agree to the reviewer's suggestion. The MiniBEST Viral RNA/DNA Extraction Kit Ver.5.0 (TaKaRa, Dalian, China; Cat No. 9766) has been used to extract the nucleic acids from the supernatant of fecal samples for many years in our laboratory. The results are very good and reach to our demand. We have ever compared the results of this kit with those of other kits from different manufactures, and confirmed the extracted results of the kit used in this study. Please see Lines 144-147 in the revised manuscript.
- Lane 195, include the RNA concentration that you used in the reaction.
Response: We agree to the reviewer's suggestion. The total nucleic acids were extracted from the clinical samples, and used to detect PSV, PKV, PTV, and EV-G directedly using the developed quadruplex RT-qPCR. We did not detect and calculate their RNA concentrations. Please see Lines 208-216 in the revised manuscript.
Results
- Table 2, in the volume column 20 µl of RNA are mentioned, however the final reaction volume is 20 µl, check and correct if necessary. Include the average concentration used for the samples.
Response: We agree to the reviewer's suggestion. “20” has been changed to “2.0”. The plasmid constructs with final concentration of 107 copies/ was used to perform RT-qPCR to obtain the optimal reaction conditions. Please see Table 2 in the revised manuscript.
- Table 5, indicate in the positive samples the average quantification value plus the standard deviation values.
Response: We agree to the reviewer's suggestion. The average quantification value plus the standard deviation values of the positive samples have been added in the notes of Table 5. Please see Lines 322-324 in the revised manuscript.
Reviewer #5
Comments and Suggestions for Authors
The reviewed manuscript is dedicated to the design and development of a multiplex PCR test for detection of 4 viruses infecting swines: porcine sapelovirus (PSV), porcine kobuvirus (PKV), porcine teschovirus (PTV), and porcine enterovirus G (EV-G). The presented assay will be indeed useful for disease control in pig farms and will help to study epidemiology of these widespread pathogens that pose a heavy burden on the industry. However, several issues listed below need to be clarified before possible publication of the manuscript.
- Page 3, lines 96–97: “, but RT-PCR has obvious shortcomings such as cumbersome operation and poor specificity.”
Response: We agree to the reviewer's suggestion. The sentence has been revised. Please see Lines 100-101.
- Authors are encouraged to provide a more detailed description of how the primers were selected. This step is crucial and defines sensitivity of PCR in terms of its ability to detect all lineages of the targeted viruses. For instance, a number of aligned genomes needs to be stated, because RNA viruses are notoriously known for their genomic instability.
Response: We agree to the reviewer's suggestion. The information has been added. Please see Lines 122-131 and Table 1 in the revised manuscript.
- Extraction of Nucleic Acid — what was the volume of the elution buffer?
Response: We agree to the reviewer's suggestion. The volume of the elution buffer has been added. Please see 147-148 in the revised manuscript.
- Construction of Standard Plasmids — were plasmid controls linearized before the usage? Efficacy of PCR with supercoiled plasmids as templates is usually lower than with linearized plasmids.
Response: We agree to the reviewer's suggestion. The supercoiled plasmids were treated with EcoRI endonuclease to obtain the linearized plasmids before use. The information has been added. Please see 161-163 in the revised manuscript.
- The abbreviation Ct is recommended to be replaced with Cq (10.1373/clinchem.2008.112797).
Response: We agree to the reviewer's suggestion. Ct has been replaced by Cq. Please see Lines 179 and 238, and Table 3 and Table 4 in the revised manuscript.
- Optimization of reaction system and reaction procedure — was it possible to reduce the concentration of the control plasmids? The designated one seems to be excessive and could lead to contamination. What was the PCR machine used in these experiments?
Response: We agree to the reviewer's suggestion. The final reaction concentration of 107 copies/μL was used as template to optimize the optimal reaction conditions. According to the results, the concentration was suitable. It did not lead to contamination. The ABI Q5 Real-Time System (Carlsbad, CA, USA) was used in this study. Please see Lines 167-168 in the revised manuscript.
- Analytical Specificity Analysis — what was the RNA concentration is samples used for the analysis of specificity?
Response: We agree to the reviewer's suggestion. The final reaction concentration of plasmid constructs as positive control were 107 copies/μL. The total nucleic acids from positive clinical samples of PSV, PKV, PTV, and EV-G were used as templates. Two microliters of total nucleic acids of each virus were used, but its concentration was not determined. Please see Lines 186-191 in the revised manuscript.
- Analytical Sensitivity Analysis — determination of LoD using plasmids for tests concerning RNA does not account for the efficacy of a reverse transcription step in RT-qPCR. Therefore, real analytical sensitivity can be different, and LoDs are recommended to determine using RNA as templates. Suitable RNA controls can be obtained after in vitro transcription of the control plasmids if the plasmid contain T7 or other similar promoter. The same consideration is valid for the repeatability analysis.
Response: We agree to the reviewer's suggestion. The recombinant standard plasmid constructs and/or cRNA are usually used to evaluate the sensitivity and repeatability of the developed RT-qPCR. In our study, the recombinant standard plasmid constructs were used. We consider it is acceptable. Please see Lines 192-207 in the revised manuscript.
- Figure 2 seems to be overloaded by curves and the presented results could be more readable if divided in several graphs.
Response: We agree to the reviewer's suggestion. The former Figure 2 has been divided in several graphs. Please see Figure 2 in the revised manuscript
- 5. Sensitivity — it seems that too many significant figures are provided for each LoD value.
Response: We agree to the reviewer's suggestion. To evaluate the LODs, the plasmid constructs were 10-fold serially diluted from 108 to 101 copies/μL (The final concentration: 107 to 100 copies/) to perform RT-qPCR. The results indicated that the LODs were 101 copies/ for all four plasmid constructs. In addition, the LODs were also assessed using Probit regression analysis to further validate the above method. The concentrations of 500, 250, 125 and 62.5 copies/reaction for p-PSV, p-PKV, p-PTV, and p-EV-G were selected as templates, and the LODs were about 140 copies/reaction. Please see Lines 267-277, Figure 3 and Figure 3, and Table 3 in the revised manuscript.
- Table 3 — SD needs to be provided for the Cq values in the “Ct Value (Average)” column.
Response: We agree to the reviewer's suggestion. SD has been added in the Table 3. Please see Table 3 in the revised manuscript.
- Table 4 — it seems that too many significant figures are provided for each value.
Response: We agree to the reviewer's suggestion. The data have been revised. Please see Table 4 in the revised manuscript.
- Assessment Results of Clinical Samples — Cq values of discordant samples would be highly welcomed, as they could hint on the reason of discrepancy between different PCR tests.
Response: We agree to the reviewer's suggestion. The Cq values for the positive samples have been added. Please see Lines 322-324 in the revised manuscript.
- Authors are encouraged to compare analytical specificity and sensitivity of the presented PCR-test with tests published by other groups and discuss possible limitations of the study.
Response: We agree to the reviewer's suggestion. The comparison of analytical specificity and sensitivity of the presented PCR-test with reference was discussed in the Section Discussion. Please see Lines 369-374 in the revised manuscript.
Best regards,
Kaichuang Shi

Reviewer 2 Report
Comments and Suggestions for Authors
In this study, Li et al. developed a quadruplex RT-qPCR assay for the simultaneous detection of PSV, PKV, PTV, and EV-G. This assay offers a rapid, sensitive, and accurate method for detecting all four viruses concurrently. Overall, the experimental design is reasonable and the results are reliable. However, there are some issues that need to be addressed.
1. The genome accession number for the reference strain must be provided.
2. The detailed description of the sample information is lacking.
3. Please ensure that the subtitles of sections 2.8 and 2.9 are consistent; verify and make any necessary modifications.
4. The sensitivity of the detection method needs to be reflected in the abstract.
Author Response

(The authors gave the same response as above.)

Reviewer 3 Report
Comments and Suggestions for Authors
The authors developed an RT-qPCR for the detection of four picornaviruses of interest in the swine industry. These viruses are primarily associated with gastrointestinal signs, such as diarrhea.
Following the design stage, recombinant pMD18-T vectors carrying each of the viral specific fragments were constructed, and the standard curves were built.
The analysis of the sensitivity and repeatability was carried out using, in addition to the four previously mentioned viruses, the genomes of other common swine viruses, including TGEV, PCV2, PRRSV, FMDV, CSFV, PRV, SIV, PoRV, ASFV, PEDV, and PDCoV.
The evaluation of the RT-qPCR technique was performed with 1,823 clinical samples obtained from Guangxi province in China. In the fine-tuning process with these clinical samples, the authors demonstrated the specificity of the designed primers, as no amplification of the genome of other common swine viruses was observed. Additionally, the study demonstrated the RT-qPCR technique's high sensitivity, with an average detection of 138 copies per microliter, accompanied by satisfactory inter- and intra-assay repeatability values.
The clinical samples were analyzed, yielding positive percentages of 15.25% (PSV), 21.72% (PKV), 18.82% (PTV), and 27.10% (EV-G). The presence of multiple viruses in various regions was determined, indicating the circulation of the viruses under study.
The study is methodically structured and written, meeting the minimum criteria for acceptance by the journal.
Minor comments:
The paper highlights the presence of principally PTV genotype 2 in China, as well as other genotypes not yet identified. The question of whether this was taken into account in the primer design was resolved, as the sequence alignment showed the name of at least 11 genotypes.
Author Response

(The authors gave the same response as above.)

Reviewer 4 Report
Comments and Suggestions for Authors
The study presented by Biao Li and collaborators describes the establishment of a quadruplex RT-qPCR for the molecular diagnosis of several Picornaviruses of interest in swine farming. The methodology used is adequately described and the findings of the study are correctly discussed.
Lane 63 and others, change the word asymptomatic to subclinical
Lane 72 and others, change the word symptoms to signs
M&M
Lane 107, it is certainly very useful to use reference strains of related viruses that infect pigs, some even from different families or genera. What about the other viruses of the Piconaviridae family that infect pigs? Why didn't you consider it? Did you perform in silico analysis of your primers and probes that include members of this viral family?
Lane 126, indicate the age range of the piglets from which the samples were obtained. Mention whether the samples came from intensive production farms or from some other type.
Lane 134, the manufacturer does not have feces on its list of recommended samples, how did you verify this?
Lane 195, include the RNA concentration that you used in the reaction.
Results
Table 2, in the volume column 20µl of RNA are mentioned, however the final reaction volume is 20 µl, check and correct if necessary. Include the average concentration used for the samples
Table 5, indicate in the positive samples the average quantification value plus the standard deviation values.
Author Response

(The authors gave the same response as above.)

Reviewer 5 Report
Comments and Suggestions for Authors
The reviewed manuscript is dedicated to the design and development of a multiplex PCR test for detection of 4 viruses infecting swines: porcine sapelovirus (PSV), porcine kobuvirus (PKV), porcine teschovirus (PTV), and porcine enterovirus G (EV-G). The presented assay will be indeed useful for disease control in pig farms and will help to study epidemiology of these widespread pathogens that pose a heavy burden on the industry. However, several issues listed below need to be clarified before possible publication of the manuscript.
- Page 3, lines 96–97: “, but RT-PCR has obvious shortcomings such as cumbersome operation and poor specificity.”
- Authors are encouraged to provide a more detailed description of how the primers were selected. This step is crucial and defines sensitivity of PCR in terms of its ability to detect all lineages of the targeted viruses. For instance, a number of aligned genomes needs to be stated, because RNA viruses are notoriously known for their genomic instability.
- Extraction of Nucleic Acid — what was the volume of the elution buffer?
- Construction of Standard Plasmids — were plasmid controls linearized before the usage? Efficacy of PCR with supercoiled plasmids as templates is usually lower than with linearized plasmids.
- The abbreviation Ct is recommended to be replaced with Cq (10.1373/clinchem.2008.112797)
- Optimization of reaction system and reaction procedure — was it possible to reduce the concentration of the control plasmids? The designated one seems to be excessive and could lead to contamination. What was the PCR machine used in these experiments?
- Analytical Specificity Analysis — what was the RNA concentration is samples used for the analysis of specificity?
- Analytical Sensitivity Analysis — determination of LoD using plasmids for tests concerning RNA does not account for the efficacy of a reverse transcription step in RT-qPCR. Therefore, real analytical sensitivity can be different, and LoDs are recommended to determine using RNA as templates. Suitable RNA controls can be obtained after in vitro transcription of the control plasmids if the plasmid contain T7 or other similar promoter. The same consideration is valid for the repeatability analysis.
- Figure 2 seems to be overloaded by curves and the presented results could be more readable if divided in several graphs.
- 5. Sensitivity — it seems that too many significant figures are provided for each LoD value.
- Table 3 — SD needs to be provided for the Cq values in the “Ct Value (Average)” column.
- Table 4 — it seems that too many significant figures are provided for each value.
- Assessment Results of Clinical Samples — Cq values of discordant samples would be highly welcomed, as they could hint on the reason of discrepancy between different PCR tests.
- Authors are encouraged to compare analytical specificity and sensitivity of the presented PCR-test with tests published by other groups and discuss possible limitations of the study.
Author Response

(The authors gave the same response as above.)

Round 2
Reviewer 5 Report
Comments and Suggestions for Authors
Many thanks to authors for their detailed comments and careful corrections of the manuscript. The most of the mentioned in first review questions have been cleared. However, a few of them still need further clarification.
- Table 3, page 13, line 355 — Ct instead of Cq.
- Currently, the main issue with the work is lack of data about sensitivity on RNA templates. If the tested viruses contained DNA genomes, the usage of control plasmid would be acceptable for assessment of analytical sensitivity. However, the efficacy of reverse transcription cannot be 100% meaning not all targeted viral RNAs will be converted into cDNA and suitable as templates for PCR. Thus, before acceptance, LoDs must be validated on RNA templates.
- The calculated LoDs for multiplex PCR are presented with 3 digits after the decimal sign. This is an excessive number of significant figures, because the real accuracy of quantitative PCR is not that high.
- Assessment Results of Clinical Samples. Authors designated Cq ranges for positive samples, but did not provide Cq values for samples that were discordant between the designed multiplex PCR and the reference PCR tests.
- Page 13, lines 365–366: “which is 1.38 to 6.58 times higher than the reference RT-qPCR” — the presented LoDs are actually lower than the LoDs of reference tests. Possibility of false-negative results due to mutations under primers and probes also needs to be discussed with possible solutions for this problem.
Author Response
The Cover Letter
March 18, 2025
Dear editor,
Our manuscript has been further revised carefully according to the reviewer's suggestions. The details were as follows.
Reviewer 5 (round 2)
Comments and Suggestions for Authors
Many thanks to authors for their detailed comments and careful corrections of the manuscript. The most of the mentioned in first review questions have been cleared. However, a few of them still need further clarification.
- Table 3, page 13, line 355 — Ct instead of Cq.
Response: We agree to the reviewer's suggestion. Cq has been revised to Ct through this manuscript. Please see Line 186, Line 245, Line 336, Line 337, Table 3, and Table 4 in the revised manuscript.
- Currently, the main issue with the work is lack of data about sensitivity on RNA templates. If the tested viruses contained DNA genomes, the usage of control plasmid would be acceptable for assessment of analytical sensitivity. However, the efficacy of reverse transcription cannot be 100% meaning not all targeted viral RNAs will be converted into cDNA and suitable as templates for PCR. Thus, before acceptance, LoDs must be validated on RNA templates.
Response: We agree to the reviewer's suggestion. In the supplementary experiments, the synthesized RNAs were used to analyze the sensitivity of the developed quadruplex RT-qPCR. The results of using RNAs are similar to the those of using plasmid constructs. Please see Lines 199-207, Lines 275-286, Figure 3, Figure 4, and Table 3 in the revised manuscript.
- The calculated LoDs for multiplex PCR are presented with 3 digits after the decimal sign. This is an excessive number of significant figures, because the real accuracy of quantitative PCR is not that high.
Response: We agree to the reviewer's suggestion. The LODs of the developed quadruplex RT-qPCR were determined through using 10-fold serially diluted synthesized viral RNAs from 1.0 × 107 to 1.0 × 100 copies/, and the results indicated the LODs of PSV, PKV, PTV,and EV-F were 1.0 × 101 copies/μL.
In addition, the mixtures of RNAs with 500, 250, 125, 62.5 copies/reaction were also used as templates for sensitivity analysis of the developed assay using Probit regression analysis, and the results showed that the LODs of PSV, PKV, PTV, and EV-G were 146.02, 143.83, 141.92, and 139.79 copies/reaction, respectively. These LODs were determined through calculation according to the methods of Probit regression analysis, and 2 digits were used in the revised manuscript.
Please see Lines 199-207, Lines 275-286, Figure 3, Figure 4, and Table 3 in the revised manuscript.
- Assessment Results of Clinical Samples. Authors designated Cq ranges for positive samples, but did not provide
Response: The clinical samples were detected using the developed quadruplex RT-qPCR in this study and the reference RT-qPCR/RT-PCR assays reported by other scientists. In these assays, the positive samples were determined as positive samples according to the Ct threshold. In our study, the samples with Ct values ≤ 36 were determined to be positive samples. In the reference assays, the samples with Ct values ≤ 36 were determined to be positive samples. The clinical samples in this study were detected using the developed assay and the reference assays, respectively, and were determined whether to be positive samples or not according to the abovementioned judgment criteria. Therefore, the Ct values for samples that were discordant between the designed multiplex PCR and the reference PCR tests were not provided. Please see Lines 311-328 in the revised manuscript.
- Page 13, lines 365–366: “which is 1.38 to 6.58 times higher than the reference RT-qPCR” — the presented LoDs are actually lower than the LoDs of reference tests. Possibility of false-negative results due to mutations under primers and probes also needs to be discussed with possible solutions for this problem.
Response: We agree to the reviewer's suggestion. The sentence “which is 1.38 to 6.58 times higher than the reference RT-qPCR” has been revised. Please see Lines 386-388 in the revised manuscript.
In addition, the possibility of false-negative results due to mutations under primers and probes has been discussed in the Section Discussion. Please see Lines 395-405 in the revised manuscript.
Best ragards,
Kaichuang Shi

Round 3
Reviewer 5 Report
Comments and Suggestions for Authors
Many thanks to authors for their detailed comments and careful corrections of the manuscript. The remained questions were cleared and the manuscript can be accepted for publication in the present form.
Author Response
March 27, 2025
Dear reviewer,
Reviewer 5 (Round 3)
Comments and Suggestions for Authors
Many thanks to authors for their detailed comments and careful corrections of the manuscript. The remained questions were cleared and the manuscript can be accepted for publication in the present form.
Response: Thanks very much for the reviewer's affirmation of the revised manuscript.
Best regards,
Kaichuang Shi
